# Synthesis, Characterization, and In Vivo Anti-Cancer Activity of New Metal Complexes Derived from Isatin-*N*(4)antipyrinethiosemicarbazone Ligand Against Ehrlich Ascites Carcinoma Cells

**DOI:** 10.3390/molecules24183313

**Published:** 2019-09-11

**Authors:** Fathy El-Saied, Bishoy El-Aarag, Tarek Salem, Ghada Said, Shaden A. M. Khalifa, Hesham R. El-Seedi

**Affiliations:** 1Department of Chemistry, Faculty of Science, Menoufia University, Shebin El-Koom 32512, Egypt; elsayedfathy139@yahoo.com (F.E.-S.); ghadasaid_2007@yahoo.com (G.S.); 2Biochemistry Division, Chemistry Department, Faculty of Science, Menoufia University, Shebin El-Koom 32512, Egypt; 3Division of Chemistry and Biotechnology, Graduate School of Natural Science and Technology, Okayama University, Okayama 7008530, Japan; 4Department of Molecular Biology, Genetic Engineering & Biotechnology Institute, University of Sadat City, Sadat City 32958, Egypt; salem_tarek@yahoo.com; 5Department of Molecular Biosciences, The Wenner-Gren Institute, Stockholm University, SE 106 91 Stockholm, Sweden; shaden.khalifa@su.se; 6Department of Experimental Cancer Medicine (ECM); Novum, 14157 Huddinge, Stockholm, Sweden; 7Pharmacognosy Group, Department of Medicinal Chemistry, Uppsala University, Biomedical Centre, Box 574, SE-751 23 Uppsala, Sweden; 8International Research Center for Food Nutrition and Safety, Jiangsu University, Zhenjiang 212013, China; 9Al-Rayan Research and Innovation Center, Al-Rayan Colleges, Medina 42541, Saudi Arabia

**Keywords:** metal complexes, isatin-*N*(4)antipyrinethiosemicarbazone, Ehrlich ascites carcinoma, tumor volume, VEGF, caspase-7

## Abstract

The current study aimed to synthesize new metal coordination complexes with potential biomedical applications. Metal complexes were prepared via the reaction of isatin-*N*(4)anti- pyrinethiosemicarbazone ligand **1** with Cu(II), Ni(II), Co(II), Zn(II), and Fe(III) ions. The obtained metal complexes **2**–**12** were characterized using elemental, spectral (^1^H-NMR, EPR, Mass, IR, UV-Vis) and thermal (TGA) techniques, as well as magnetic moment and molar conductance measurements. In addition, their geometries were studied using EPR and UV–Vis spectroscopy. To evaluate the in vivo anti-cancer activities of these complexes, the ligand **1** and its metal complexes **2**, **7** and **9** were tested against solid tumors. The solid tumors were induced by subcutaneous (SC) injection of Ehrlich ascites carcinoma (EAC) cells in mice. The impact of the selected complexes on the reduction of tumor volume was determined. Also, the expression levels of vascular endothelial growth factor (VEGF) and cysteine aspartyl-specific protease-7 (caspase-7) in tumor and liver tissues of mice bearing EAC tumor were determined. Moreover, their effects on alanine transaminase (ALT), aspartate transaminase (AST), albumin, and glucose levels were measured. The results revealed that the tested compounds, especially complex **9**, reduced tumor volume, inhibited the expression of VEGF, and induced the expression of caspase-7. Additionally, they restored the levels of ALT, AST, albumin, and glucose close to their normal levels. Taken together, our newly synthesized metal complexes are promising anti-cancer agents against solid tumors induced by EAC cells as supported by the inhibition of VEGF and induction of caspase-7.

## 1. Introduction

Cancer is considered one of the most devastating life-threatening disease worldwide. It is characterized by uncontrolled cell growth and the rapid spread of abnormal cells [1]. Angiogenesis is a vital regulator of tumor growth, spread and metastasis [2]. Tumor angiogenesis, which is the development of new blood vessels, is regulated by the production of angiogenic stimulators such as vascular endothelial growth factor (VEGF). VEGF is a key regulatory factor in the prognosis of various cancers; therefore, the inhibition of VEGF production is an alternative therapeutic approach for cancer treatment [3,4,5,6].

Chemotherapy, along with surgery, radiation, hormone therapy, and immune therapy, is used to treat cancer [7]. Despite the success of controlling the tumor growth in several cases, the incidence and prevelance of cancer is still dramtically increasing and calling for new chemotherapeutics to be introduced to the pharmaceutical industry and the market [8]. Antipyrine (2,3-dimethyl-1-phenyl-5-pyrazolone) and its derivatives possess a diverse array of potentially useful biological properties [9]. Similarly, thiosemicarbazones have demonstrated significant biological applications, as antitumor [10], antifungal [11], antiviral [12], antibacterial [13] and antimalarial [14] agents. Previous investigations involving the synthesis, characterization and biological evaluation of iron(III), cobalt(II), cobalt(III), nickel(II), zinc(II), and copper(II) complexes of 4-formylantipyrine *N*(4)-substituted thiosemicarbazones have been reported [15,16].

Based on these results and as a continuation of our ongoing interest in compounds with antitumor activities, the present study was carried out to synthesize and characterize metal complexes of isatin-*N*(4)antipyrinethiosemicarbazone and to evaluate their anti-tumor, anti-angiogenic and apoptotic activities against solid tumors induced by Ehrlich ascites carcinoma (EAC) cells in mice.

## 2. Results and Discussion

All the prepared metal complexes, shown in Figure 1, are stable at room temperature, non-hygroscopic and insoluble in most organic solvents and water, but they are completely soluble in dimethylformamide (DMF) and dimethylsulfoxide (DMSO). The elemental analyses, physical data, are listed in Table 1. The elemental analyses confirmed that complexes **2**–**6**, **8** and **10** are formed in a 1:1 molar ratio of the metal ion and ligand, whereas complexes **9**, **11** and **12** are formed in a 1:2 molar ratio of the metal ion and the ligand. Complex **7** is a binuclear complex.

### 2.1. ^1^H-NMR Spectra

The ^1^H-NMR spectra of the ligand **1** and the zinc complex **11** are illustrated in the Appendix A. The data shows three separate singlets at 12.819, 11.219 and 10.115 ppm, which can be attributed to the thiosemicarbazide N-N(2)H, the indole N-H moiety and the thiosemicabazide CS-N(4)-H, respectively [17]. Multiple signals appear in the 7.702–6.876 ppm region, and these can be attributed to the aromatic protons of the indole and phenyl rings of antipyrine. The other signals for the ligand at 3.068 (s, 3H) and 2.135 (s, 3H) correspond to the N-CH_3_ and C-CH_3_ of the antipyrine moiety, respectively [15]. The ^1^H-NMR spectrum of complex **11** does not show a signal corresponding to N-N(2)H of thiosemicarbazide, suggesting the loss of the thiol proton, and the binding ligand in its thiol form. The spectrum displays signals attributable to the indole N-H moiety, the CS-N(4)-H of thiosemicabazide and all the other characteristic signals at the same positions as those of the free ligand.

### 2.2. Mass Spectra

The mass spectra of ligand **1** and complexes **7**, **8**, **11**, and **12** are shown in the Appendix A. The molecular ion peak at *m*/*z* 406.27 amu in the spectrum of the ligand **1** is consistent with the molecular weight of its proposed structure. Moreover, the fragmentation pattern shows ion peaks at *m*/*z* 363.47, 291.15, 276.68, 261.69, 223.71, 204.07, 203.30, 145.64, 130.57, 118.18 and 76.3 amu corresponding to C_19_H_17_N_5_OS, C_13_H_17_N_5_OS, C_12_H_13_N_5_OS, C_12_H_13_N_4_OS, C_9_H_10_N_4_OS, C_9_H_6_N_3_OS, C_11_H_13_N_3_O, C_8_H_5_N_2_O, C_8_H_5_NO, C_7_H_5_NO and C_6_H_5_ fragments, respectively.

The mas spectrum of complex **7** (F.W. = C_40_H_38_N_12_O_6_S_2_Co_2_Cl_2_) shows a molecular ion peak at *m*/*z* 999.5 amu, consistent with a molecular weight of 998.8 g/mol, which is the molecular weight of the complex after the loss of one molecule of hydration water (F.W. = C_40_H_34_N_12_O_4_S_2_Co_2_Cl_2_). The spectrophotometer at hand cannot detect molecular ions with molecular weights greater than 1000 g/mol. The fragment ion peaks at *m*/*z* 868.9 and 463.9 amu correspond to the formulas C_40_H_34_N_12_O_4_S_2_Co and C_20_H_17_N_6_O_2_Co.

The mass spectrum of complex **8** (C_22_H_30_N_6_O_9_SCo, F.W. = 612.5) displays a molecular ion peak at *m*/*z* 612.85 (100%) amu, confirming the suggested structure of the complex. Moreover, the spectrum shows several peaks, including peaks at *m*/*z* 576.23 (94.07), 561.30 (61.93), 541.85 (6.86), and 523.22 (39.20), corresponding to the formula C_22_H_26_N_6_O_7_SCo, C_22_H_24_N_6_O_6_SCo, C_22_H_22_N_6_O_5_SCo, and C_22_H_20_N_6_O_4_SCo, respectively, due to the sequential loss of two molecules of water of hydration, another water of hydration, a final water of hydration, and one coordinated water molecule, respectively.

The mass spectrum of complex **11** (C_40_H_38_N_12_O_6_S_2_Zn, F.W. = 911.50) displays a molecular ion peak at *m*/*z* 911.95 (1.19), consistent with the suggested structure.The mass spectrum of complex **12** (C_40_H_38_N_12_O_5_S_2_FeCl_3_, F.W. = 992.50) exhibits a molecular ion peak at *m*/*z* 992.50 amu, confirming the putative molecular weight of the complex. Moreover, the spectrum shows several ion peaks, including peaks at 973.50 (1.90), 938.30 (12.96), 869.80 (8.27), and 464.40 (2.48) corresponding to the formula C_40_H_36_N_12_O_4_S_2_FeCl_3_, C_40_H_38_N_12_O_5_S_2_FeCl_3_, C_40_H_38_N_12_O_5_S_2_FeCl_2_, C_40_H_38_N_12_O_5_S_2_Fe and C_20_H_18_N_6_O_2_SFe, which are attributable to the losses of one water of hydration, an ionisable chloride, two coordinated chloro ligands and one ligand, respectively.

### 2.3. The Molar Conductance

The molar conductance values of the metal complexes in DMF (10^−3^ M) were illustrated in Table 1. The results showed that none of the complexes except **12** are electrolytes [16], indicating that the anions in all these complexes are directly bound to the metal ion. Complex **12** gave a molar conductance of 74.5 Ω^−1^cm^2^mol^−1^ attributable to a 1:1 electrolyte, and indicating the presence of a chloride ion in the outer coordination sphere [18]. The high value of molar conductance of complex **6** may be associated with the partial displacement of the perchlorato ligand by DMF [15].

### 2.4. Infrared (IR) Spectra

The IR spectra of the ligand **1** and its metal complexes **2**–**12** were shown in Appendix A. The most diagnostic infrared spectral bands are illustrated in Table 2; the infrared spectrum of the ligand **1** displays bands at 3442, 3348, 3287, 3251, and 3141 attributable to ν(N-H) absorptions characteristic of –NH groups [19,20]. The spectrum also displays bands at 1738, 1645, 1688, 1623 and 1595 cm^−1^. The first band can be assigned to ν(C=O) of the isatin moiety [19,20], and the second band can be assigned to ν(C=O) of the antipyrine moiety. The latter three bands are associated with ν(C=N) [21]. The appearance of multiple bands characteristic of ν(C=N) can be explained by the formation of tautomers secondary to the presence of the isatin carbonyl group and C=S with the adjacent –NH groups. The absorption band of ν(C=S) (thioamide IV) observed at 881 cm^−1^ [15] and a weak band observed at 2650 cm^−1^, assigned to ν(C-SH), supported the thione-thiol tautomerization expected from the C=S group with an adjacent proton [22].

The binding mode of the ligand in each metal complex has been deduced by comparing the IR spectrum of the ligand with that of the corresponding metal complex. In the infrared spectra of all the complexes, the band characteristic of the ν(C=N) of the isatin moiety is shifted to lower wave numbers compared to that of the free ligand, indicating the coordination of C=N group with the metal via the nitrogen atom in all complexes. In the IR spectra of the metal complexes, except for those of **7** and **8**, the bands corresponding to the C=O of the isatin moiety appeared at lower wavenumbers compared to that of the ligand as a result of the participation of the carbonyl oxygen atom of this group in the coordination. This band did not shift in the infrared spectra of complexes **7** and **8**, indicating that C=O is not involved in the coordination.

In the infrared spectra of the complexes, except **9**, **10** and **12**, the absorption band is attributable to the ν(C=O) of the antipyrine moiety appeared at the same position as that of the ligand, indicating that this group only participates in coordination to the metal ion in complexes **9**, **10** and **12**. In the infrared spectra of all complexes, except **9** and **12**, the band corresponding to ν(C=S) appeared at lower wavenumbers (808–848 cm^−1^) compared to that of the free ligand, indicating that the C-S group takes part in the coordination. The appearance of a band at 1622–1560 cm^−1^, which is assignable to υ(–N=C-S-), accompanies the shift in the ν(C-S) band. The appearance of the later band indicates the participation of the thiol form in coordination rather than the thion form, as suggested by the ^1^H-NMR spectrum of complex **11**. The ν(C=S) bands in the infrared spectra of complexes **9** and **12** were at the same position as that of the free ligand, indicating that the C=S moiety does not participate in coordination in complexes **9** and **12**.

Complexes **9** and **12** are composed of two ligand molecules that bind to one metal ion, and their infrared spectra showed that the two ligand molecules have different binding modes. In the infrared spectra of the complexes **9** and **12**, each exhibits four bands at 1735–1737, 1700–1660, 1645 and 1620 cm^−1^, attributable to the uncoordinated ν(C=O) of the isatin moiety, the coordinated ν(C=O) of the isatin moiety, the uncoordinated ν(C=O) of the antipyrine moiety and the coordinated ν(C=O) of the antipyrine moiety, respectively. Moreover, each of the spectra of complexes **9** and **12** displays two bands corresponding to the ν(C=N) of the isatin moiety at 1595 and 1520 cm^−1^, which are assignable to the free uncoordinated C=N and the coordinated C=N, respectively. The above results illustrate that one of the ligand molecules behaves as a neutral, bidentate ligand and coordinates via the C=N and C=O groups of the isatin moiety and that the second molecule also behaves as a neutral, bidentate ligand but coordinates via the C=O and N-H groups of the antipyrine moiety.

The spectra of acetato complexes **5**, **8** and **10** show two absorption bands at 1620–1615 and 1372–1333 cm^−1^, assigned to υ(C=O) and υ(C-O), respectively. The difference between these two bands is approximately 287−243 cm^−1^, which indicates that the acetate in these complexes coordinates to the metal ion as a monodentate ligand [23]. The monodentate coordination of a perchlorate ligand lowers the symmetry of the system (T_d_→C_3v_). The modes of the free perchlorate that were initially degenerated (υ_3_ and υ_4_) are both split into two bands [24]. The infrared spectrum of the perchlorate [Cu(ClO_4_)L]·2H_2_O complex showed that both the υ_3_ band and υ_4_ band are split into two bands, at 1118 and 1085 cm^−1^ and at 694 and 631 cm^−1^, respectively, indicating that the perchlorate coordinates to the metal ion as a monodentate ligand. The infrared spectrum of complex **4**, [Cu(NO_3_)L]·H_2_O showed two bands at 1462 and 1382 cm^−1^, attributable to a monodentate nitrato ligand [25]. The infrared spectra of the hydrated complexes display a broad band at 3491–3388 cm^−1^ from ν(OH) of the water molecule [26]. The IR spectra of all the complexes display a new band at 591–498 cm^−1^, assigned to ν(M-N) [27,28,29], and the spectra of all the metal complexes except **7** and **8** show a new band at 693–589 cm^−1^, assigned to υ(M-O) [19,20,21,22,23,24,25,26,27,28,29].

### 2.5. Magnetic and Electronic Spectra

The room-temperature magnetic moments (μ_eff_ BM per metal atom) and electronic spectral bands for the metal complexes in the solid state are described in Table 3. The data showed that the electronic absorption spectrum of ligand **1** exhibits bands at 268 and 280 nm, attributable to intra-ligand π→π* electronic transitions [15,30]. These bands do not change significantly upon complex formation [31].

The somewhat broad band at 369 nm is assignable to the intra-ligand n→π* electronic transitions associated with the combination of the azomethine function of the thiosemicabazone, the thione function of the thiosemicabazone and the heterocyclic moiety [15,32]. In the spectra of the metal complexes, this band generally shifts to a higher energy as a result of the participation of the azomethine and/or thiol groups in coordination [33,34]. In the spectra of the metal complexes, new bands were observed in the 357–530 nm region due to S(π)→metal(II,III) charge-transfer processes [35,36], and the spectrum of bromo complex **3** showed a Br→metal(II) charge-transfer band [37]. Cl→metal(II,III) [38,39,40] and O→metal(II,III) [35] charge-transfer bands were generally observed at approximately 333 nm and were obscured by the stronger intra-ligand transfer bands. Copper(II) complexes **2**–**6** showed magnetic moments of 1.76–1.87 BM. These values are close to the spin-only value for one unpaired spin (~1.73 BM) and indicate that there is no molecular association between the copper(II) ions in the square planar environment [34,41,42]. The electronic absorption spectra of Cu(II) complexes **2**–**5** showed two or three bands at 870–660, 740–631 and 675–580 nm, assignable to the ^2^B_1_g(d_x2−y2_)→^2^A_1g_(d_z2_), ^2^B_1_g(d_x2−y2_)→^2^B_2g_(d_xy_) (υ_2_) and ^2^B_1_g(d_x2−y2_)→^2^Eg(d_xz_,d_yz_) (υ_3_) transitions, suggesting a square planar geometry around the copper(II) ion. Complex **6** displays a broad band at 790 nm that could be attributed to the abovementioned transitions in the ligands in a square planar geometry around the copper(II) ion [43,44,45].

Cobalt(II) complex **7** gave a magnetic moment of 3.1 BM, which is considerably less than what is expected for high-spin tetrahedral cobalt(II). This unexpected value could be explained by the occurrence of antiferromagnetic exchange through the chloro bridge linking the two cobalt(II) ions [46,47]. Complex **8** gave a magnetic moment of 3.7 BM per metal ion, which is consistent with high-spin tetrahedral cobalt(II) complexes [48]. The electronic spectra of complexes **7** and **8** displayed a broad band with several features at approximately 675 nm, which is attributable to the ^4^A_2_→^4^T_1_(P) (v_3_) transition in a tetrahedral geometry around the Co(II) ion [15].

The nickel(II) complexes were found to be paramagnetic, which ruled out square planar configurations. The magnetic moments of nickel(II) complexes **9** and **10** are 2.98 and 3.06 BM, respectively, corresponding to two unpaired electrons [49], suggesting geometries other than square planar. The electronic spectra of Ni(II) complexes **9** and **10** showed bands at 950, 690-675 nm and 560−567 nm, assignable to the ^3^A_2g_(**F**)→^3^T_2g_(**F**)(υ_1_), ^3^A_2g_(**F**)→^3^T_1g_(**F**)(υ_2_) and ^3^A_2g_(**F**)→^3^T_1g_(**P**) transitions, respectively, in an octahedral environment [50,51,52]. The υ_2/_υ_1_ ratios for these complexes are 1.41 and 1.37, which are below the usual range of 1.5–1.75. These ratios suggest a tetragonally distorted octahedral geometry around the nickel(II) ions [51,53].

Iron(III) complex **12** gave a magnetic moment of 5.95 BM per metal ion, corresponding to a high-spin d^5^ configuration. The electronic spectrum of complex **12** showed strong bands at 391, 450, 530 and 584 nm. The first three bands can be assigned to ligand-to-metal charge-transfer (LMCT) processes. The last band at 584 nm is broad and of relatively high intensity. The relatively high intensity of this band can be attributed to overlap with a low-lying ligand charge-transfer band. This is because in the case of iron(III), the d-d transitions are considered forbidden because they take place between ions with different multiplicities; examples of such transitions include the ^6^A_1_g→^4^T_1g_ (G), ^6^A_1_g→^4^T_2g_ (G) and ^6^A_1_g→^4^E_g_ (G) or ^4^E_g_(D) transitions in tetragonally distorted octahedral Fe(III) environments [54]. The electronic spectrum of the zinc(II) complex showed bands at 396, 440, 491, and 520(sh), which can be attributed to ligand-to-metal (L→M) charge-transfer processes.

### 2.6. EPR Spectra of the Copper(II) Complexes

The EPR spectra of the Cu(II) complexes are illustrated in the Appendix A. The EPR data of Cu(II) complexes **2**–**6** in the polycrystalline state at 298 K are listed in Table 4. The EPR spectra of the Cu(II) complexes have low g_1_ and g_av_ values, indicating that the bonds have considerable covalent characteristics [18,55]. The EPR spectra of the Cu(II) complexes other than complex **5** ([Cu(L)(OAc)]·2H_2_O) show rhombic-type distortion with three g values, g_1_ = 2.189, 2.198, 2.287, and 2.25; g_2_ = 2.121, 2.079, 2.121, and 2.12; and g_3_ = 2.057, 1.989, 1.977, and 2.027, respectively. The rhombic symmetry of these complexes is attributable to the bulkiness of the ligand and to the unequal bond lengths of the four Cu-S, Cu-N, Cu-O, and Cu-X bonds (X=Cl, Br, NO_3_ or ClO_4_) [55]. The g tensors, which follow g_1_ ˃ g_2_ ˃ g_3_ ˃ 2, are indicative of complexes with a d_(x2−y2)_ ground state. This result is supported by the value of R (R = (g_2_ − g_1_/(g_3_ − g_2_) < 1 (= 0.94 − 0.72)), confirming a d_(x2−y2)_ ground state with covalent bonding character [56,57].

The EPR spectral data of complex **5** demonstrate anisotropic signals with g_‖_ > g_⊥_ > 2.0023, which are indicative of a d_(x2−y2)_ ground state with axial symmetry, resulting in a ^2^B_1g_ ground state that is known for Cu^2+^ complexes [58,59]. These g values are consistent with Cu^2+^-complex **5** having a square planar geometry [60,61]. The geometric parameter G, which is a measure of the exchange interaction between the copper centres in the solid state, was calculated using the equation G = (g_‖_ − 2.0023)/(g_⊥_ − 2.0023) for axial and rhombic compounds, g_‖_ = g_1_ and g_⊥_ = g_2_ + g_3_/2. If G ˃ 4, the exchange interaction is negligible, and if it is less than 4, considerable exchange interactions exist [58,62]. Complexes **3**, **4**, and **5** have G values ˃ 4, indicating that exchange interactions are negligible, while for complexes **2** and **6**, G ˂ 4, suggesting considerable exchange interactions between the Cu(II) centres. Kivelson and Neiman reported that the g_‖_-values of Cu^2+^ complexes can be used to determine the nature of the copper-ligand bonding. If the g_‖_-value is smaller than 2.3, the environment is essentially covalent, while if the value is larger than 2.3, the environment is essentially ionic [63]. For all the synthesized complexes, g_‖_ < 2.3, indicating that there is considerable covalent character in the copper-ligand bonds.

According to [45,64,65], the orbital reduction factor (k) and its parallel (k_‖_) and perpendicular (k_⊥_) components were calculated according to the following equations: k_‖_^2^ = (g_‖_ − 2.0023)∆_2_/8λ_0_, k_⊥_^2^ = (g_⊥_ − 2.0023) ∆_1_/2 λ_0_, k^2^ = (K_‖_^2^ + 2_⊥_^2^)/3 where λ_o_ is the spin orbit coupling (−828 cm^−1^) of the free Cu(II) ion, and Δ_1_ and Δ_2_ are the electronic transitions ^2^B_1g_→^2^B_2g_ (d_x2−y2_→d_xy_) and ^2^B_1g_→^2^E_g_ (d_x2−y2_→d_xz_d_yz_), respectively. The calculated k_‖_^2^, k_⊥_^2^ and k^2^ values in Table 1 showed that for complexes 3–6, k_‖_ ˃ k_⊥_, indicating the presence of significant out-of-plane π-bonding [66] and confirming that ^2^B_1g_ is the ground state for these complexes. For complex 2, k_‖_ ˂ k_⊥_, indicating the presence of significant in-plane π-bonding [67]. Moreover, when the environment is ionic, k = 1, but if this value is less than 1, the environment is covalent. These compounds showed k values lower than one, which is strong evidence of covalent character, which is in agreement with the results obtained from the g_‖_ values [65,68]. None of the solid-state EPR spectra of the complexes displayed hyperfine structure, confirming the presence of intermolecular interactions, but these interactions were not strong enough to overcome the dipolar interactions [69].

### 2.7. Thermal Analysis

The TGA spectra of metal complexes are shown in the Appendix A. The TGA results of solid complexes **2**, **3**, **5** and **7**–**12** are listed in Table 5. The results show an agreement with the formulae suggested by the analytical, spectral and magnetic data. A general decomposition pattern, in which the complexes decompose in two or three stages, was observed. The first stage is the loss of waters by hydration at 29–160 °C; the second stage of decomposition corresponds to the loss of coordinated water molecules at 80–175 °C; complex **8** showed a third decomposition stage that is a result of overlapping processes including the loss of Cl^−^, Br^−^ or OAc^−^ anions; the loss the ligands to form metal oxides or metal sulfides mixed with carbon residues; and the oxidation of carbon to carbon dioxide leaving metal oxides or sulfides.

### 2.8. Biological Activities

#### 2.8.1. Effect of Ligand 1 and Its Metal Complexes **2**, **7** and **9** on Solid Tumor Volume

Mice bearing solid tumors were treated SC with 0.2 mM of ligand **1** and its metal complexes **2**, **7** and **9**. The treatment was started seven days after EAC implementation in mice and was continued for 14 consecutive days. Tumor volume was measured after 24 h from the last dose of treatment. As shown in Table 6, treatment of solid tumor-bearing mice with the ligand **1** resulted in a significant reduction in tumor volume with inhibition of (36.6%) as compared with tumor-bearing mice treated with DMSO. Metal complexes **2**, **7** and **9** exhibited potent effects on reduction of the volume of solid tumor. The recorded inhibition percents were 66.5%, 70.37% and 75.53%, respectively, as compared with tumor-bearing mice treated with the vehicle.

#### 2.8.2. Effect of Ligand **1** and Its Metal Complexes **2**, **7** and **9** on the Biochemical Analysis of Serum

As shown in Table 7, EAC cells induced a significant reduction in albumin and glucose and a significant elevation in alanine aminotransferase (ALT) and aspartate aminotransferase (AST) levels (*p* < 0.001) in comparison with normal mice (normal control group). Conversly, ligand **1** and its metal complexes **2**, **7** and **9** exhibited significant increase of albumin and glucose levels and decrease ALT and AST levels relative to tumor-bearing mice treated with DMSO (Table 7).

An increase in glucose catabolism and a decrease in glucose synthesis in the presence of cancer cells were reported [70,71]. Therefore, the decrease in serum glucose in mice bearing solid tumor may be due to the tumors exhibit high rate of glycolysis. This is a way of stimulating the body into a constant state of gluconeogenesis which is accompanied with the reduction in serum albumin [72,73]. Treatment of tumor-bearing mice with ligand **1** and its metal complexes, especially complex **7**, restored serum glucose, albumim, ALT and AST levels towards the standard levels when compared with the tumor-bearing mice treated with the vehicle.

#### 2.8.3. Effect of Ligand **1** and Its Metal Complexes **2**, **7** and **9** on the Expression of Vascular Endothelial Growth Factor (VEGF) in Tumor and Liver Tissues of EAC Mice

VEGF is a significant angiogenic factor released during the tumor formation to stimulate the angiogenesis process. Usage of VEGF antagonists is one of the essential anti-angiogenic therapies for cancer treatment [74]. Therefore, the effect of ligand **1** and its metal complexes **2**, **7** and **9** on VEGF expression was studied in tumor and liver tissues of EAC mice. Results as presented in Figure 2A,B illustrated that VEGF is highly expressed in tumor and liver tissues of mice bearing solid tumor compared to normal mice.

The increasing in the VEGF expression level in mice bearing solid tumors was related to the presence of EAC cells. This finding is consistent with previously reported results which concluded that EAC cells has been found to exhibit strong VEGF expression level [75]. Moreover, cancer usually enhances VEGF expression through tumor aggression, therefore, VEGF is involved in cancer pathology [76]. Conversely, the treatment with ligand **1** and its metal complexes **2**, **7** and **9** downregulated the expression of VEGF in tumor (Figure 2A) and in liver tissues (Figure 2B) of mice bearing solid tumor in comparison to mice bearing solid tumor treated with DMSO. Complex **9** exhibited the most potent anti-VEGF activity among the tested compounds. These results revealed that the anti-tumor activities of the tested compounds might be due to their ability to inhibit VEGF.

#### 2.8.4. Effect of Ligand **1** and Its Metal Complexes **2**, **7** and **9** on the Expression of Cysteine Aspartyl-Specific Protease-7 (Caspase-7) in Tumor and Liver Tissues of EAC Mice

Apoptosis is a programmed cell death that is caused by caspases that target cysteine aspartyl residues in target proteins [77]. Defects in apoptosis mechanisms can induce tumor pathogenesis [78], for example, the activation of caspases by the signal transduction pathway can lead to irreversible apoptosis through protein degradation [79]. Therefore, it is important to design and develop new agents that act as caspases activators to treat cancers [80,81]. Consequently, the effect of ligand **1** and its metal complexe **2**, **7** and **9** on the stimulation of caspase-7 was examined in tumor and liver tissues of EAC mice.

As shown in Figure 3A,B, the induction of solid tumors is associated with low expression of caspase-7 in tumor and liver tissues of mice bearing solid tumor compared to normal mice. Our finding was compatible with the reported data demonstrating the declined expression of caspase-3 in mice bearing EAC cells [82]. On the other hand, the treatment with ligand **1** and its metal complexes **2**, **7** and **9** upregulated the expression of caspase-7 in tumor (Figure 3A) and in liver tissues (Figure 3B) of mice bearing EAC tumor in comparison to mice bearing solid tumor treated with DMSO. 

Complex **9** followed by complex **7** exhibited the most potent apoptotic activity among the tested compounds. Caspase-7 is a critical mediator of apoptosis and the overexpression of it induced apoptosis of cancer cells [83]. Therefore, these results revealed that the anti-tumor activities of the tested compounds might be due to their ability to stimulate caspase-7.

## 3. Materials and Methods

### 3.1. Chemistry

#### 3.1.1. Preparation of *N*(4)-antipyrinylthiosemicarbazide

The preparation of S-methylantipyrine carbodithioate was illustrated in Scheme 1. In details, 4-Aminoantipyrine (40.6 g, 0.2 mol) was added in small portions to an aqueous KOH solution (11.2 g, 100 mL), and the mixture was maintained below 5 °C using ice. Refrigerator-cooled CS_2_ (11.36 cm^3^, 0.2 mol) was then added dropwise to the stirred solution over l h while the solution was maintained below 10 °C. To the resulting yellow solution, MeI (11.68 cm^3^, 0.2 mol) dropwise was added over 2 h while the temperature was maintained below 10 °C. During the addition of MeI, the yellow colour of the mixture gradually diminished, and a white solid powder was formed. The crystals of S-methyl-antipyrinecarbodithioate were collected by vacuum filtration and washed with cold H_2_O. Because of the noxious odour of this solid (m.p. 193 ± 2 °C), it was used immediately without purification.

The preparation of N(4)-antipyrinylthiosemicarbazide was presented in Scheme 2. *S*-Methylantipyrinecarbodithioate (29.3 g, 0.1 mol) dissolved in EtOH (30 mL) was added to hydrazine monohydrate (3.2 mL, 0.1 mol). The solution was heated at reflux until the evolution of methyl mercaptan was almost complete. The release of methyl mercaptan was detected based on the yellow colour it imparts to paper moistened with Pb(OAc)_2_ placed at the mouth of the condenser. The reaction time for this step was 8 h. The resulting white solid crystals, *N*(4)-antipyrinyl-thiosemicarbazide, were collected by vacuum filtration, washed with cold i-PrOH and dried (m.p. 203 ± 2 °C).

#### 3.1.2. Preparation of Isatin *N*(4)-antipyrine-thiosemicarbazide **1** (ligand)

Isatin N(4)-antipyrine-thiosemicarbazide **1** (ligand) was prepared by mixing equimolar amounts of isatin and *N*(4)-antipyrine-thiosemicarbazide in anhydrous EtOH with a few drops of concentrated H_2_SO_4_ at reflux (80 °C) for 2 h as shown in Scheme 3.

Ligand **1**: Yield (87%), m.p. = 210–212 °C; color: yellow, Elemental analysis for C_20_H_18_N_6_O_2_S, (F.W. = 406.): Found (calcd) %C 59.14 (59.00), %H 5.15(4.43), %N 20.91(20.66), %S 7.85(7.87)); IR (KBr, cm^−1^), 3442, 3348, 3287, 3251, 3141ν(NH), 1738 ν(C=O)_isatin_, 1645 ν(C=O)_antipyrine_, 1688, 1623, 1595 ν(C=N), 881 ν(C=S); UV-Vis. (Nujol mulls) (nm) 268, 280, 369 nm, ^1^H-NMR (MHz, DMSO-d_6_, δ, ppm): 12. 819 (s, 1 H, thiosemicabazide N-NH), 11.219 (s, 1H, indole N-H), 10.115 (s, 1H, CS-N-H), 7.702–6.876. (m, multiple, H aromatic protons), 3.068 (s, 3 H, N-CH_3_), 2.135 (s, 3 H, CH_3_). GC–MS: *m*/*z* (relative abundance); 406.27, 363.47 (27.15), 291 (100), 276.68 (72.92), 261.69 (28.97), 223.71 (90.05), 204.07 (22.73), 203.3 (26.0), 145.64 (51.57), 130.57 (28.60), 118.18 (78.92), 67.30 (18.68).

#### 3.1.3. Preparation of the Metal Complexes

The metal complexes of the ligand were prepared by mixing a hot ethanolic solution (30 mL) of the metal salt (CuCl_2_·2H_2_O, CuBr_2_, Cu(NO_3_)_2_·3H_2_O, Cu(OAc)_2_·H_2_O, Cu(ClO_4_)_2_, CoCl_2_·6H_2_O, Co(OAc)_2_·4H_2_O, NiCl_2_·6H_2_O, Ni(OAc)_2_·4H_2_O, ZnCl_2_ and FeCl_3_·6H_2_O) with the appropriate amount of a hot (75 °C) ethanolic solution of the ligand to form a 1:1 L/M (ligand/metal) solution. The reaction mixture was then refluxed for approximately 2 h. The obtained precipitates were separated by filtration, washed several times with ethanol and then diethyl ether, and then dried under vacuum over anhydrous CaCl_2_. Due to the poor solubility of these complexes in most organic solvents and water, all attempts to prepare a single crystal of the ligand or any of the metal complexes were unfortunately unsuccessful.

*[Cu(L)Cl]·H_2_O* (**2**): Yield (79%), m.p. = 238–240 °C; color: dark brown, Λ_M_: 10.0 ohm^−1^ cm^2^ mol^−1^, Elemental analysis for C_20_H_19_N_6_O_3_SCuCl., (F.W. = 522.6): Found (calcd) %C 46.0(46.0), %H 4.1(3.64), %N 16.3(16.10), %S 6.88(6.13): IR (KBr, cm^−1^), 3395 ν(H_2_O), 3161, 3100 ν(NH), 1702 ν(C=O)_isatin_, 1645 ν(C=O)_antipyrine_, 1616, 1539 ν(C=N), 808 ν(C=S), 632 ν(Cu-O), 498 ν(Cu-N) UV-Vis. (Nujol mulls) (nm) 310, 325 n→π*, 395, 460, 525 Charge transfer, 600, 672, 725 d→d, µ_eff_ (B.M) = 1.83.

*[Cu(L)Br]·3H_2_O* (**3**): Yield (89%), m.p. = 250 °C; color: brown, Λ_M_: 37.9 ohm^−1^ cm^2^ mol^−1^, Elemental analysis for, C_20_H_23_N_6_O_5_SCuBr, (F.W. = 602.5): Found (calcd) %C 38.81(39.83), %H 3.54(3.82), %N 15.02(13.94), %S 5.03(5.31) IR (KBr, cm^−1^), 3446 ν(H_2_O), 3106ν(NH), 1695 ν(C=O)_isatin_, 1643 ν(C=O)_antipyrine_, 1622, 1578 ν(C=N), 806 ν(C=S), 614 ν(Cu-O), 504 ν(Cu-N); UV-Vis. (Nujol mulls) (nm) 325 n→π*, 380, 445, 500 charge transfer, 580, 631, 660 d→d, µ_eff_ (B.M) = 1.81.

*[Cu(L)(NO_3_)]·H_2_O* (**4**): Yield (80%), m.p. = 247 °C; color: brown, Λ_M_: 34.3 ohm^−1^ cm^2^ mol^−1^, Elemental analysis for, C_20_H_19_N_7_O_6_SCu, (F.W. = 548.5): Found (calcd) %C 43.4(43.5), %H 3.29(3.50), %N 18.0(17.9), %S 6.0(5.8); IR (KBr, cm^−1^), 3425ν(H_2_O), 3166ν(NH), 1703 ν(C=O)_isatin_, 1645 ν(C=O)_antipyrine_, 1620, 1539 ν(C=N), 1382,1462 ν(NO3), 808 ν(C=S), 638 ν(Cu-O), 586 ν(Cu-N); UV-Vis. (Nujol mulls) (nm) 310 n→π*, 390, 455, 520, 574 charge transfer, 675, 720 d→d, µ_eff_ (B.M) = 1.78.

*[Cu(L)OAc]·2H_2_O* (**5**): Yield (74%), m.p. = 212–216 °C; color: brown, Λ_M_: 11.9 ohm^−1^ cm^2^ mol^−1^, Elemental analysis for, C_22_H_24_N_6_O_6_SCu, (F.W. = 563.5): Found (calcd) %C 46.12(46.90), %H 4.82(4.30), %N 15.13 (14.9), %S 5.1(5.7); IR (KBr, cm^−1^), 3417 ν(H_2_O), 3237ν(NH), 1680 ν(C=O)_isatin_, 1645 ν(C=O)_antipyrine_, 1560, 1525 ν(C=N), 1615, 1333 ν(OAc), 847 ν(C=S), 595 ν(Cu-O), 503 ν(Cu-N); UV-Vis. (Nujol mulls) (nm) 310 n→π*, 395, 430, 460, 560 charge transfer, 671,740(br), 870 d→d, µ_eff_ (B.M) = 1.79.

*[Cu(L)ClO_4_]·2H_2_O* (**6**): Yield (91%), m.p. = 250–252 °C; color: brown, Λ_M_: 45.50 ohm^−1^ cm^2^ mol^−1^; Elemental analysis for, C_20_H_21_N_6_O_8_SCuCl, (F.W. = 604.5): Found (calcd) %C 38.92(39.70), %H 3.75(3.48), %N 14.67(13.91), %S 5.18(5.29); IR (KBr, cm^−1^), 3428 ν(H_2_O), 3325, 3170ν(NH), 1628 ν(C=O)_isatin_, 1645 ν(C=O)_antipyrine_, 1568, 1516ν(C=N), 1118,1085, 694,631 ν(ClO4), 838 ν(C=S), 693 ν(Cu-O), 505 ν(Cu-N); UV-Vis. (Nujol mulls) (nm) 310 n→π*, 357, 390, 480(br) charge transfer, 790(br) d→d, µ_eff_ (B.M) = 1.76.

*[Co(L)Cl]_2_·2H_2_O* (**7**): Yield (66%), m.p. = 258–260 °C; color: brown, Λ_M_: 17.9 ohm^−1^ cm^2^ mol^−1^, Elemental analysis for C_40_H_38_N_12_O_6_S_2_Co_2_Cl_2_, (F.W. = 1034.8): Found (calcd) %C 47.0(46.38), %H 4.17(3.7), %N 17.0(16.21), %S .4(6.18), IR (KBr, cm^−1^), 3388 ν(H_2_O), 3150 ν(NH), 1734 ν(C=O)_isatin_, 1645 ν(C=O)_antipyrine_, 1600, 1515 ν(C=N), 841 ν(C=S), 585 ν(Co-N); GC–MS: *m*/*z* (relative abundance); 999.5 (22.46), 869.45 (15.13), 464.94 (4.00); UV-vis. (Nujol mulls) (nm) 318, 345 n→π*, 390, 450(br) charge transfer, 600, 675, 700 d→d, µeff (B.M) = 3.1.

*[Co(L)OAc(H_2_O)]·4H_2_O* (**8**): Yield (75%), m.p. ˃ 300 °C; color: brown, Λ_M_: 4.6 ohm^−1^ cm^2^ mol^−1^, Elemental analysis for C_22_H_30_N_6_O_9_SCo, (F.W. = 612.5): Found (calcd) %C 42.96(43.10), %H 5.3(4.89),%N 14.20(13.71), %S 4.87(5.22); IR (KBr, cm^−1^), 3491, 3422 ν(H_2_O), 3196ν(NH), 1734 ν(C=O)_isatin_, 1645 ν(C=O)_antipyrine_, 1590, 1545 ν(C=N), 1619, 1372 ν(OAc), 841 ν(C=S), 540 ν(Co-N); GC–MS: *m*/*z* (relative abundance); 612.85 (100), 576.23 (94.07, 561.30 (61.93), 541.85 (6.86), 523.22 (39.20); UV-Vis. (Nujol mulls) (nm) 320 n→π*, 390, 440(br), 530(br) charge transfer, 600, 675, 780d→d, µ_eff_ (B.M) = 3.7

*[Ni(HL)_2_Cl_2_]·4H_2_O* (**9**): Yield (84%), m.p. = 254–255 °C; color: brown, Λ_M_: 13.5 ohm^−1^ cm^2^ mol^−1^, Elemental analysis for C_40_H_44_N_12_O_8_S_2_ NiCl_2_., (F.W. = 1014.9): Found (calcd) %C 46.84(47.3), %H 4.94(4.33), %N 16.95(16.55), %S 5.89(6.30); IR (KBr, cm^−1^), 3441 ν(H_2_O), 3280 ν(NH), 1735, 1660 ν(C=O)_isatin_, 1645, 1620 ν(C=O)_antipyrine_, 1595(sh), 1520(sh) ν(C=N), 880 ν(C=S), 648 ν(Ni-O), 591 ν(Ni-N); UV-VIS. (Nujol mulls) (nm) 320 n→π*, 396, 420, 530 charge transfer, 560(s), 690(w), 950(br) d→d, µ_eff_ (B.M) = 2.98.

*[Ni(L)OAc]_2_·4H_2_O* (**10**): Yield (70%), m.p. ˃ 300 °C; color: brown, Λ_M_: 2.60 ohm^−1^ cm^2^ mol^−1^, Elemental analysis for C_44_H_48_N_12_O_12_S_2_ Ni_2_, (F.W. = 1017.4): Found (calcd) %C 47.76(47.24), %H 4.19(4.3), %N, 15. 60(15.03), %S 6.71(5.78); IR (KBr, cm^−1^), 3445 ν(H_2_O), 3310, 3200 ν(NH), 1664 ν(C=O)_isatin_, 1620 ν(C=O)_antipyrine_, 1580, 1540 ν(C=N), 1620, 1342 ν(OAc), 829 ν(C=S), 590 ν(Ni-O), 503 ν(Ni-N); UV-Vis. (Nujol mulls) (nm) 305, 315 n→π, 395, 450, 530 charge transfer, 567(s), 670(sh). 950(br) d→d, µ_eff_ (B.M) = 3.06.

*[Zn(L)_2_]·2H_2_O* (**11**): Yield (69%), m.p. ˃ 300 °C; color: dark yellow, Λ_M_: 1.40 ohm^−1^ cm^2^ mol^−1^, Elemental analysis for C_40_H_38_N_12_O_6_S_2_ Zn, (F.W. = 911.50): Found (calcd) %C 51.9(52.6), %H 3.7 (4.2), %N 18.8 (18.42), %S 6.64 (7.0); IR (KBr, cm^−1^ 3413 ν(H_2_O), 3223ν(NH), 1700 ν(C=O)_isatin_, 1643 ν(C=O)_antipyrine_, 1609, 1570 ν(C=N), 815 ν(C=S), 612 ν(Zn-O), 580 ν(Zn-N): ^1^H-NMR (MHz, DMSO-*d*_6_, δ, ppm): 11.22 (s, 1H, indole N-H), 11.008 (s,1H, CS-N-H), 7.704–6.879. (m, multiple, C-H aromatic), 3.111 (s, 3 H, N-CH3), 2.174 (s, 3 H, CH3); GC–MS: *m*/*z* (relative abundance); 911.5 (1.19) UV-Vis. (Nujol mulls) (nm) 319 n→π, 396, 440, 491, 520(sh).

*[Fe(HL)_2_Cl_2_]Cl·H_2_O* (**12**): Yield (78%), m.p. = 210–212 °C; color: dark green, Λ_M_: 74.50 ohm^−1^ cm^2^ mol^−1^, Elemental analysis for C_40_H_38_N_12_O_5_S_2_FeCl_3_, (F.W. = 992.50): Found (calcd) %C 47.62(48.3), %H 3.9(3.8), %N 16.95 (16.91), %S 5.6 (6.44), IR (KBr, cm^−1^), 3429(s,br) ν(H_2_O), 3280 ν(NH), 1737, 1700 ν(C=O)_isatin_, 1645, 1621 ν(C=O)_antipyrine_, 1610, 1576 ν(C=N), 880 ν(C=S), 589 ν (Fe-O), 512 ν(Fe-N) GC–MS: *m*/*z* (relative abundance); 992.5 (11.39), 974.50 (1.90), 938.30 (12.96), 869.80 (8.27), 464.40 (2.48); UV-Vis. (Nujol mulls) (nm) 300, 340 n→π, 391, 450, 530(sh) charge transfer, 567(s), 584 d→d, µ_eff_ (B.M) = 5.95.

#### 3.1.4. Instrumentation and Measurements

Elemental microanalyses [C, H, and N] were conducted in the Micro Analytical Unit. The IR spectra were acquired using KBr discs in a Spirit model FT-IR spectrophotometer (Shimadzu, Thermo Scientific, Glasgow, UK). The ^1^H-NMR spectra of the ligand and zinc complex were recorded in DMSO-*d*_6_ on a Gemini 200 NMR spectrophotometer (Varian, Palo Alto, CA, USA) at 300 MHz. Mass spectra were collected using a model ISQLT direct probe controller with a single quadrupole mass analyser (Thermo Scientific, Glasgow, UK) using Thermo X-calibur software (Xcalibur 4.1). Solid-state electronic spectra were recorded in Nujol mulls using a Shimadzu UV-Vis 1800 spectrophotometer (Thermo Scientific, Glasgow, UK). The molar conductance of a 1 × 10^−3^ M solution of the complexes in *N*,*N*-dimethylformamide (DMF) was measured at 25 °C with a Bibby MCl-type conductometer (Chelmsford, UK). The resistance was measured in ohms and the molar conductivities were calculated according to the following equation: ΛM = VxKxg/Mw* Ω where ΛM = molar conductivity (Ω-1cm2mol-1), V = volume of the complex solution (mL), K = cell constant (0.92 cm-1), Mw = molecular weight of the complex, g = weight of the complex (g), and Ω = resistance (Ω).

The solid-state electron paramagnetic resonance (EPR) spectra of the copper(II) complexes were recorded with an X-band EMX spectrometer (Bruker, Berlin, Germany) using a standard rectangular cavity of ER 4102 operating at 9.5 GHz with 100 kHz modulation at 298 K. Diphenyl picryl hydrazide (DPPH) was used as a g-marker to calibrate the spectra. The thermal analyses (TGA) were carried out in air using a Shimadzu DT-30 thermal analyser. Magnetic susceptibilities were measured at 27 °C using the modified Gouy method with a Johnson Matthey balance and were calculated using the following equation:
$$\mathrm{Meff}=2.84\sqrt{x_{m}^{\mathrm{corr}}\cdot T}$$


### 3.2. Biological Evalution

#### 3.2.1. Preparation of Solutions of the Complexes

Stock solutions (1 mM) of the newly synthesized complexes were prepared by dissolving each complex in dimethylsulfoxide (DMSO)/H_2_O, and the solutions were stored at 4 °C. Only ligand 1 and complexes **2**, **7** and **9** were successfully solubilized, allowing their antitumor activities to be determined.

#### 3.2.2. Experimental Animals

Adult female Swiss albino mice (6–7 weeks old), weighing 20–22 g, were purchased and housed under standard ventilation and light cycle conditions. They were given food and water ad libitum. The current study protocol was approved by the Ethical Committee for Laboratory Animals of Science Faculty Menoufia University (No.: ECLA-SFMU-18015).

#### 3.2.3. Tumor Cell Line

The Ehrlich ascites carcinoma (EAC) cell line was purchased from the National Cancer Institute (NCI, Cairo, Egypt). The EAC cell line was maintained in the peritoneal cavity of adult female Swiss albino mice by serial intraperitoneal transplantation of 2.5 × 10^6^ cells weekly.

#### 3.2.4. Induction of Solid Tumor and Animals Grouping

The solid tumor was induced by subcutaneously (SC) injecting 1 × 10^−6^ EAC cells into the right thigh of the lower limb of mice [84,85]. Seven days after tumor inoculation, the tumor volume became palpable. The mice were randomly divided in separate cages into six groups:Group 1 (normal control): mice without any treatment.Group 2 (vehicle group): mice bearing solid tumors treated with DMSO/H_2_O.Group 3: mice bearing solid tumor and treated subcutaneously with 0.2 mM of ligand **1**.Group 4: mice bearing solid tumor and treated subcutaneously with 0.2 mM of complexe **2**.Group 5: mice bearing solid tumor and treated subcutaneously with 0.2 mM of complexe **7**.Group 6: mice bearing solid tumor and treated subcutaneously with 0.2 mM of complexe **9**.

The doses were selected based on the LD_50_ values of the complexes (data not shown). The treatment was started seven days after EAC implementation and was continued for 14 consecutive days.

#### 3.2.5. Blood and Tissue Samples Collection

At the end of the experiment, mice were sacrificed. Blood samples were collected from jugular vein puncture into chilled non-heparinized tubes, which were centrifuged at 40× *g* for 10 min at 4 °C. The sera were frozen at −20 °C for future measurements. The livers and solid tumors were removed and cut into pieces, and the individual pieces were fixed in paraffin blocks for immunohistochemical assays.

#### 3.2.6. Assessment of Tumor Volume (TV)

After 24 h from the last dose of treatment, mice were sacrificed and the solid tumors were carefully removed. The volume of the tumors was assessed [86] using the following formula: TV (mm^3^) = 4π (A/2)^2^ × (B/2), where (A) is the minor tumor axis and (B) is the major tumor axis.

#### 3.2.7. Biochemical Analysis of the Serum

To determine the effect of ligand **1** and its metal complexes **2**, **7** and **9** on the serum biochemical profile. The levels of alanine transaminase (ALT), aspartate transaminase (AST), albumin and glucose were determined using a Beckman autoanalyzer apparatus, Brea, CA, USA).

#### 3.2.8. Immunohistochemical Examination of VEGF and Caspase-7 in Solid Tumors and Liver Tissues

The effect of ligand **1** and its metal complexes **2**, **7** and **9** on the expression of VEGF and caspase-7 in solid tumor tissues and in liver tissues of mice bearing solid tumors was deremined by immunohistochemical methods. In details, the sections of tumor and liver tissues in paraffin were dewaxed, hydrated, and immersed in antigen retrieval solution (EDTA solution, pH 8). The samples were treated with 0.3% hydrogen peroxide and protein block and then incubated with anti-VEGF (Santa Cruz, Dallas, TX, USA, 1:100 dilution) and anti-caspase-7 (Lab Vision, Thermo Fischer Scientific, Glasgow, UK; ready to use) at 4 °C overnight. The slides were rinsed three times with PBS and incubated with anti-mouse IgG secondary antibodies (EnVision + System HRP; Dako, SanJose, CA, USA) for 30 min at room temperature. The samples were visualized with diaminobenzidine commercial kits (Liquid DAB+Substrate Chromogen System; Dako) and finally counterstained with Mayer’s haematoxylin. In the negative control, the primary antibody was replaced with normal mouse serum. The positive cells were counted from seven high-power fields of 400×. The labelling index (%) was determined by dividing the number of positive cells by the total cells × 100 [87,88].

### 3.3. Statistical Analysis

Data are expressed as mean ± SEM and the statistical comparison between groups were analyzed using one-way ANOVA followed by Tukey post hoc test. A *p* of < 0.05 value was considered statistically significant.

## 4. Conclusions

In summary, new metal complexes derived from isatin-N-(4)antipyrinethiosemicarbazone ligand **1** were synthesized and characterized. Also, their potential anti-cancer activities against solid tumors induced by EAC cells in mice were investigated. The results revealed that the tested complexes significantly reduced the volume of solid tumors. This reduction was associated with an increase in the level of caspase-7 and a dramatic decrease in the VEGF level. The tested complexes are promising anti-cancer agents those exert their activities via the induction of apoptosis and the inhibition of angiogenesis, leading to reduction of the tumor size.

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
