# Peer review of "Synthesis, Characterization, and In Vivo Anti-Cancer Activity of New Metal Complexes Derived from Isatin-N(4)antipyrinethiosemicarbazone Ligand Against Ehrlich Ascites Carcinoma Cells"

_molecules, 2019, doi:10.3390/molecules24183313_

Round 1
Reviewer 1 Report
The authors synthesized and characterized metal complexes derived from Isatin-N(4)antipyrinethiosemicarbazone ligand. The complexes were tested also for their in vivo biological activity against soli tumor. The manuscript can be accepted for publication in Molecules, after major revision. The authors should revise the manuscript according to the following comments.
The quality of the figure 1 is too low.
The figures 2, 3 and 4 should be moved to the supplementary materials. Moreover the quality of them is too low.
The authors should give the conductance values of all the complexes
The ir spectrum should be uploaded in the supplementary materials
The UV, EPR and TGA should be uploaded in the supplementary materials
In the part " Effect of ligand 1 and its metal complexes 2, 7 and 9 on solid tumor volume" the authors should mention the used concentration, the dose, as well as the period that they measure the tumor volume
How the authors comment the increase of the expression of VEGF in tumor when they used DMSO
Please explain in the text why the authors choose for the biological activity the complexes 2, 7, and 9.
The authors should explain more clearly the results of effect of the complexes on the expression of caspase-7 .
Author Response
Point 1: The quality of the figure 1 is too low.
Response 1: The quality of figure 1, chemical structure of metal complexes 2-12, was improved in the revised manuscript.
Point 2: The figures 2, 3 and 4 should be moved to the supplementary materials. Moreover, the quality of them is too low.
Response 2: Figures 2 (H NMR spectra of ligand), 3 (H NMR spectra of complex 11) and 4 (mass spectra of ligand 1 and its metal complexes 2-12) were moved to the supplementary materials. Also, their qualities were enhanced.
Point 3: The authors should give the conductance values of all the complexes
Response 3: The molar conductance values of the metal complexes were illustrated in Table 1 in the revised manuscript.
Point 4: The ir spectrum should be uploaded in the supplementary materials
Response 4: The IR spectra were uploaded in the supplementary materials.
Point 5: The UV, EPR and TGA should be uploaded in the supplementary materials
Response 5: The UV, EPR and TGA spectra were uploaded in the supplementary materials.
Point 6: In the part " Effect of ligand 1 and its metal complexes 2, 7 and 9 on solid tumor volume" the authors should mention the used concentration, the dose, as well as the period that they measure the tumor volume.
Response 6: The used concentration, the dose, as well as the period at which the tumor volume was measured were added in section 2.8.1 in the revised manuscript as the following:
Mice bearing solid tumor were treated subcutaneously with 0.2 mM of ligand 1 and its metal complexes 2, 7 and 9. The treatment was started seven days after EAC implementation in mice and was continued for 14 consecutive days. Tumor volume was measured after 24 h from the last dose of treatment.
Point 7: How the authors comment the increase of the expression of VEGF in tumor when they used DMSO.
Response 7: The role of DMSO was used as a vehicle to dissolve the ligand and its metal complexes. Tumor bearing mice treated with vehicle (DMSO group) was taken as a tumor model group. The increase in the expression of VEGF in tumor model group was explained in section 2.8.3 in the revised manuscript as the following:
The increasing in the VEGF expression in mice bearing solid tumor was related to the presence of EAC cells. This finding is consistent with previously reported results which concluded that EAC cells has been found to exhibit strong VEGF expression level [80]. Moreover, cancer usually enhances VEGF expression through tumor aggression, therefore, VEGF is involved in cancer pathology [81].
Additionally, to remove the confusion and make Fig. 2 (former Fig. 5) and Fig. 3 (former Fig. 6) more readable, these figures were revised and the term DMSO was replaced by EAC-DMSO.
Point 8: Please explain in the text why the authors choose for the biological activity the complexes 2, 7, and 9.
Response 8: The reason that the authors selected the complexes 2, 7 and 9 for the biological activity is that, these compounds were successfully solubilized at the used concentrations, allowing their antitumor activities to be determined. This explanation was added in section 3.2.1 in the revised manuscript.
Point 9: The authors should explain more clearly the results of effect of the complexes on the expression of caspase-7.
Response 9: More details explanations for the results of the complexes on the expression of caspase-7 were added in section 2.8.4 in the revised manuscript.
Reviewer 2 Report
This manuscript describes the synthesis and characterisation of a series of metal complexes with one or two isatin-N(4)antipyrinethiosemicarbazone ligands. In vivo studies in mice bearing tumours are also reported for selected compounds, these results are promising. Overall the manuscript is worth of publication once the following critical points have been addressed.
A number of characterisation techniques have been used, however, without an X-ray crystal structure of the metal complexes their connectivity as depicted in Fig.1. cannot be confirmed. I would urge the authors to try and report some X-ray crystal structures of the metal complexes, otherwise the authors should edit the text throughout the manuscript to add caution to the proposed structures. Clear NMR spectra should be provided and the peaks need to be assigned. The mass spectra peaks need to be assigned. UV-Vis spectra should be provided for each compound. Why were complexes 2, 7 and 9 selected for the biological studies? What was the rationale? The authors should provide more details on the in vivo experiments. How were the compounds administered? Positive control compounds should be included in the in vivo experiments if possible.Author Response
Point 1: A number of characterization techniques have been used, however, without an X-ray crystal structure of the metal complexes their connectivity as depicted in Fig.1. cannot be confirmed. I would urge the authors to try and report some X-ray crystal structures of the metal complexes, otherwise the authors should edit the text throughout the manuscript to add caution to the proposed structures.
Response 1: The authors apologize as they can’t perform X-ray crystal structure of the metal complexes. Therefore, the authors added a caution in section 3.1.3. in the revised manuscript that involved the inability to perform X-ray crystal structure of the metal complexes as the following:
Due to the poor solubility of these complexes in most organic solvents and water, all attempts to prepare a single crystal of the ligand or any of the metal complexes were unfortunately unsuccessful.
Point 2: Clear NMR spectra should be provided and the peaks need to be assigned.
Response 2: Clear NMR spectra with assigned peaks were added and moved to the complementary materials.
Point 3: The mass spectra peaks need to be assigned.
Response 3: The mass spectra peaks were assigned and the spectra were moved to the complementary materials.
Point 4: UV-Vis spectra should be provided for each compound.
Response 4: The UV-Vis spectra of all complexes were added in the complementary materials.
Point 5: Why were complexes 2, 7 and 9 selected for the biological studies? What was the rationale?
Response 5: The reason that the authors selected the complexes 2, 7 and 9 for the biological activity is that, these compounds were successfully solubilized at the used concentrations, allowing their antitumor activities to be determined. This explanation was added in section 3.2.1 in the revised manuscript.
Point 6: The authors should provide more details on the in vivo experiments.
Response 6: Further details on the in vivo experiments were added in sections 3.2.4, 3.2.6, 3.2.7, and 3.2.8 in the revised manuscript.
Point 7: How were the compounds administered?
Response 7: The compounds were administrated via subcutaneous (SC) route into the thigh of the lower limb of mice. The administration route was mentioned in section 3.2.4 in the revised manuscript.
Point 8: Positive control compounds should be included in the in vivo experiments if possible.
Response 8: Thank you for your comment. In the current study, the authors don’t use positive control compound, however, this will be taken into account in the future study.
Round 2
Reviewer 1 Report
The authors have answered the comments. The manuscript can be accepted now in the Molecules
Author Response
All authors thank the reviewer for taking the time and effort necessary to provide the acceptance of our paper.
Reviewer 2 Report
The authors have made some minimal changes to the manuscript. However there some major concerns remaining.
The NMR spectra have not been assigned properly - the chemical shifts on the NMR spectra do not match the chemical shifts in the written form - this is highly worrying.
The NMR spectra are unclear. The peak multiplicities cannot be seen.
The mass spectra have not been assigned. The spectra should be annotated with the relevant peaks highlighted.
The UV spectra of the complexes looks very strange. The concentration used is probably too high - the spectra should be re-done with low concentrations.
Author Response
All authors thank the reviewer for taking the time and effort necessary to provide such insightful guidance. Also, we agree with the comments of reviewer 2 and found them very helpful.
Below, we address each criticism and the comments made by the referee 2 individually, and explain how we have modified the manuscript to address the concerns that were expressed. We kindly ask that you consider the revised manuscript for publication.
Comments and Suggestions for Authors
Point 1: The NMR spectra have not been assigned properly - the chemical shifts on the NMR spectra do not match the chemical shifts in the written form - this is highly worrying.
Response 1: The NMR spectra were checked and revised. The peaks of each spectrum were consistent with those written in the text of the revised manuscript.
Point 2: The NMR spectra are unclear. The peak multiplicities cannot be seen.
Response 2: The NMR spectra were completely improved and the peak multiplicities became obvious in the revised complementary materials.
Point 3: The mass spectra have not been assigned. The spectra should be annotated with the relevant peaks highlighted.
Response 3: The metal complexes possessed complicated structures. The mass spectra explained the molecular mass of the synthesized metal complexes. The molecular mass of each complex was clear in each spectrum and compatible with that reported in the revised manuscript.
Point 4: The UV spectra of the complexes looks very strange. The concentration used is probably too high - the spectra should be re-done with low concentrations.
Response 4: The authors were tried many concentrations to get the most suitable UV spectra. The used concentration in the current study showed the most satisfied UV spectra. The broadness in the peaks may be due to the charge transfer transitions.
Thank you once again for your time and consideration.